# Brain-Inspired Spatio-Temporal Associative Memories for Neuroimaging Data Classification: EEG and fMRI

**DOI:** 10.3390/bioengineering10121341

**Published:** 2023-11-21

**Authors:** Nikola K. Kasabov, Helena Bahrami, Maryam Doborjeh, Alan Wang

**Affiliations:** 1Knowledge Engineering and Discovery Research Innovation, School of Engineering, Computer and Mathematical Sciences, Auckland University of Technology, Auckland 1010, New Zealand; helena.bahrami@aut.ac.nz (H.B.); maryam.gholami.doborjeh@aut.ac.nz (M.D.); 2Intelligent Systems Research Center, University of Ulster, Londonderry BT48 7JL, UK; 3Institute for Information and Communication Technologies, Bulgarian Academy of Sciences, 1113 Sofia, Bulgaria; 4Computer Science and Engineering Department, Dalian University, Dalian 116622, China; 5Auckland Bioengineering Institute, University of Auckland, Auckland 1010, New Zealand; 6Knowledge Engineering Consulting Ltd., Auckland 1071, New Zealand; 7Core & Innovation, Wine-Searcher, Auckland 0640, New Zealand; 8Royal Society Te Apārangi, Wellington 6011, New Zealand; 9Research Association New Zealand (RANZ), Auckland 1010, New Zealand; 10Faculty of Medical and Health Sciences, University of Auckland, Auckland 1010, New Zealand; 11Centre for Brain Research, University of Auckland, Auckland 1010, New Zealand

**Keywords:** spatio-temporal associative memory, STAM, neuroimaging data, spiking neural networks, NeuCube, EEG, fMRI, neuroimage classification

## Abstract

Humans learn from a lot of information sources to make decisions. Once this information is learned in the brain, spatio-temporal associations are made, connecting all these sources (variables) in space and time represented as brain connectivity. In reality, to make a decision, we usually have only part of the information, either as a limited number of variables, limited time to make the decision, or both. The brain functions as a spatio-temporal associative memory. Inspired by the ability of the human brain, a brain-inspired spatio-temporal associative memory was proposed earlier that utilized the NeuCube brain-inspired spiking neural network framework. Here we applied the STAM framework to develop STAM for neuroimaging data, on the cases of EEG and fMRI, resulting in STAM-EEG and STAM-fMRI. This paper showed that once a NeuCube STAM classification model was trained on a complete spatio-temporal EEG or fMRI data, it could be recalled using only part of the time series, or/and only part of the used variables. We evaluated both temporal and spatial association and generalization accuracy accordingly. This was a pilot study that opens the field for the development of classification systems on other neuroimaging data, such as longitudinal MRI data, trained on complete data but recalled on partial data. Future research includes STAM that will work on data, collected across different settings, in different labs and clinics, that may vary in terms of the variables and time of data collection, along with other parameters. The proposed STAM will be further investigated for early diagnosis and prognosis of brain conditions and for diagnostic/prognostic marker discovery.

## 1. Introduction

Memory is referred to as the brain’s ability to recall experiences or information that is encountered or learned previously. If this information is recalled using only partial inputs, we refer to it as Associative Memory (AM) [1,2]. There are three main types of memory in the brain, namely sensory memory, short-term or working memory, and long-term memory, which function in different ways. However, each of these types is manifested through brain activities in space (areas of the brain) and time (spiking sequences), stored as connection weights and recalled always with only partial input information in time and space. AM in the brain is always spatio-temporal.

Humans can learn and understand many categories and objects from spatio-temporal stimuli by creating a spatial and temporal association between them. Inspired by the human brain capability, AM has been introduced to the machine learning field to memorize information and retrieve it from partial or noisy data. For example, neural network models for associative pattern recognition were proposed by J. Hopfield [3] and B. Kosko [4]. In 2019, Haga and Fukai [5] introduced a memory system for neural networks based on an attractor network, which is a group of connected nodes that display patterns of activity and tend towards certain states. They applied the concept of excitatory and inhibitory nodes to their proposed network to mimic the role of the hippocampus in balancing networks to form new associations. The work above is related to vector-based data (e.g., static images) and not to spatio-temporal data. None of them relate to spatio-temporal data and more specifically, to neuroimaging (NI) data.

Spatio-temporal associative memory (STAM) is defined here as a system that is trained for classification or prediction on all available spatio-temporal variables and data and recalled only on part of the spatial or/and temporal components.

The idea of using a brain-inspired and brain-structured spiking neural network (SNN) as a spatio-temporal associative memory (STAM) was first introduced in [6] as part of the NeuCube SNN architecture, but the main concepts and definitions of STAM were introduced in [7], where a NeuCube model, trained on complete spatio-temporal data, creating spatio-temporal patterns in its connections, was recalled when only partial spatial- or/and temporal information was provided as inputs.

In this paper, we introduced a general model of STAM for the classification of Neuroimaging (NI) data and then applied it for the development of STAMs on EEG and fMRI spatio-temporal data. The paper is organized as follows. Section 2 presents the background concepts of spiking neural networks (SNN), NeuCube, and STAM on NeuCube [7]. Section 3 presents a STAM-NI as a general NI classification model, while Section 4 presents a STAM-EEG model and Section 5 presents a STAM-fMRI classification model. Section 6 offers discussions about using the STAM-NI framework across bioengineering applications, including multimodal neuroimaging data, and also what are the next challenges in the development and the use of STAM as new AI techniques for the future.

## 2. SNN, the NeuCube Framework, and the STAM on the NeuCube Concept

### 2.1. Spiking Neural Networks (SNN)

Spiking neural networks (SNN) are biologically inspired ANNs where information is represented as binary events (spikes), similar to the event potentials in the brain, and learning is also inspired by principles in the brain. SNNs are also universal computational mechanisms [8,9]. Learning in SNN relates to changes in connection weights between two spatially located spiking neurons over time (Figure 1) so that both “time” and “space” are learned in the spatially distributed connections.

A well-known unsupervised learning paradigm inspired by the Hebbian learning principle is spike-time dependent plasticity (STDP) [8], in which the synaptic weights are adjusted based on the temporal order of the incoming spike (pre-synaptic) and the output spike (post-synaptic). STDP is expressed in Equation (1), where $\tau_{+}$ and $\tau_{-}$ are time parameters and $A_{+}$ and $A_{-}$ refer to temporal synaptic adjustment.
(1)$$W\left(t_{pre}-t_{post}\right)=\left\{\begin{array}{c} A_{+}\;exp\left(\frac{t_{pre}-t_{post}}{\tau_{+}}\right)\;\;\;\;\;\;\;\;\;\;\;\;if\;t_{pre}<t_{post},\\ A_{-}\;exp\left(-\frac{t_{pre}-t_{post}}{\tau_{-}}\right)\;if\;t_{pre}>t_{post}, \end{array}\right.$$

Many computational models and architectures have been developed with the use of SNN (see for a review [9]). One of them, NeuCube [6,10,11] has been used for the proposed STAM-NeuCube model and also for the STAM-NI, STAM-EEG, and STAM-fMRI models developed here.

### 2.2. The NeuCube Framework

The NeuCube architecture is depicted in Figure 2 [6]. It consisted of the following functional modules:Input data encoding module.3D SNN reservoir module (SNNcube) that is designed according a spatial brain template [12,13,14].Output function (classification) module, such as deSNN [11].Gene regulatory network (GRN) module (optional).Parameter optimization module (optional).

### 2.3. The STAM on NeuCube Concept

The main thrust of the proposed [7] STAM on NeuCube concept is that, since a SNNcube learns functional pathways of spiking activities represented as structural pathways of connections when only a small initial part of input data is entered, the SNN will ‘synfire’ and ‘chain-fire’ learned connection pathways [15] to reproduce learned functional pathways as polychronisation of neuronal clusters [16]. Some studies defined the state of a SNN as a dynamic chaotic attractor [17] that can be reached with the partial input information. In [18,19] polychronous neuronal groups are studied that are activated from partial inputs.

Spatio-temporal input data was first encoded into spike sequences and then spatio-temporal patterns of these sequences were learned in a SNNcube of the NeuCube framework that was structured according to a spatial template representing spatial information of the modeled data. For brain data, templates such as Talairach [12], MNI [13], personal MRI [14], etc., can be used. For multisensory streaming data modeling, the location of the sensors is used [9]. Connections are created and strengthened in the SNNcube through STDP learning. Once data is learned, the SNNcube retains the connections as a long-term memory.

To validate a STAM model, several types of accuracy tests were introduced in [7]:Temporal association accuracy: validating the full model on partial temporal data of the same variables.Spatial association accuracy: validation of the full model on full or partial temporal data, but on a subset of variables.Temporal generalization accuracy: validation of the full model on partial temporal data of the same variables or a subset of them, but on a new data set.Spatial generalization accuracy: validation of the full model on full or partial temporal data and a subset of variables, using a new data set.

Based on the general STAM-NeuCube concept, here we developed a specific STAM for NI data, called STAM-NI and then applied it for the development of STAM-EEG and STAM-fMRI, demonstrated on case study NI data.

## 3. The Proposed STAM-NI Classification Model and Its Mathematical Description

Spatio-temporal NI data are collected from specific brain locations over time. It is important to incorporate both the spatial and temporal information from the NI data across all measured variables over time in order to capture a meaningful pattern from the data in a computational model.

SNN and the brain-inspired NeuCube architecture have proved to be efficient in learning spatio-temporal NI data and capturing meaningful spatio-temporal patterns from the data [9]. The challenge now is how to utilize this feature for the development of STAM for the classification of NI data.

The following procedures and mathematical equations describe the proposed STAM-NI classification framework:(1)Spatial information from the NI data, e.g., 3D location of electrodes, was used to structure a SNNcube and to define the locations of the input neurons to map the NI variables. Suitable brain templates were used for this purpose [12,13,14].(2)Every spatio-temporal NI sequence, measured as a variable Vi, was encoded into a spike sequence using some of the existing methods [9]. This is illustrated in Figure 3.(3)The encoded sequences of all NI variables V were used to train a SNNcube in unsupervised mode using the STDP rule (Equation (1)), creating a connectionist structure. Before training, the SNNcube was initialized with the use of the small-world connectivity model, where neuron *a* was connected to other neuron *b* with a probability *P_a,b_* that depended on the closeness of the two neurons. The closer they are (the smaller the distance between them *D_a,b_*)*,* the higher the probability of connecting them (Equation (2)).
(2)$$p_{a,b}=C\times e^{-D^{2}{}_{a,b}/\lambda^{2}}$$*λ* is a parameter.(4)The trained SNNcube was recalled (activated) by all NI spatio-temporal data samples, one by one, using all variables as in step 3. For every sample Pi, a state Si of the SNNcube was defined during the propagation of the input sequence. The state Si was defined as a sequence of activated neurons Ni1, Ni2, …, Nil over time, that was used to train a deSNN classifier in a supervised mode [11], forming an l-element vector Wi of connection weights of an output neuron Oi assigned to the class of the input sequence Pi. For the supervised learning in the deSNN classifier, Equations (3) and (4) were used:
W_j,i_ = α Mod^order(j,i)^(3)
ΔW_j,i_ (t) = e_j_(t) D(4)where Mod is a modulation factor defining the importance of the order of the spike arriving at a synapse j of output neuron Oi, e_j_(t) = 1 if there is a consecutive spike at synapse j at time t during the presentation of the learned pattern by the output neuron I, and (−1) otherwise. In general, the drift parameter D can be different for ‘up’ and ‘down’ drifts, and α is a parameter.(5)When a new input sequence Pnew is presented, either as a full sequence in time and/or space (number of input variables) or as a partial one for STAM, a new SNNcube state Snew was learned as a new output neuron Onew. Its weight vector Wnew is compared with the weight vectors of the existing output neurons for classification tasks using the k-nearest neighbor method. The new sample Pnew was classified based on the pre-defined output classes of the closest, according to the Euclidean distance, output weight vectors Wi (Equation (5)).
Class (Pnew) = Class (Pi), if |Wnew − Wi| < |Wnew − Wk|,(5)for all output neurons Ok.(6)The temporal and spatial association and generalization accuracy were calculated.

## 4. The Proposed STAM-EEG Classification Method and Experimental Case Study

### 4.1. The Proposed STAM-EEG Classification Method


i.Defining the spatial and temporal components of the EEG data for the classification task, e.g., EEG channels and EEG time series data.ii.Designing a SNNcube that is structured according to a brain template suitable for the EEG data (e.g., Talairach, or MNI, etc.).iii.Defining the mapping in the input EEG channels into the SNNcube 3D structure (see Figure 4a as an example of mapping 14 EEG channels in a Talairach-structured SNNcube).iv.Encoding data and training a NeuCube model to classify complete spatio-temporal EEG data, having K EEG channels measured over a full-time T.v.Analyse the model through cluster analysis, spiking activity, and the EEG channel spiking proportional diagram (see for example Figure 4b,c).vi.Recall the STAM-EEG model on the same data and same variables but measured over time T1 < T to calculate the classification temporal association accuracy.vii.Recall the STAM-EEG model on K1 < K EEG channels to evaluate the classification spatial association accuracy.viii.Recall the model on the same variables, measured over time T or T1 < T on new data to calculate the classification temporal generalization accuracy.ix.Recall the NeuCube model on K1 < K EEG channels to evaluate the classification spatial generalization accuracy using a new EEG dataset.x.Evaluate the K1 EEG channels as potential classification EEG biomarkers for an early diagnosis or prognosis according to the problem at hand.


### 4.2. Experimental Results

The experimental EEG data consisted of 60 recordings of 14 EEG channels of a subject who was moving a wrist: up (class 1), straight (class 2), and down (class 3). The data included 20 samples for each class, each sample measured at 128 time points used to discretize 1000 ms signal. First, a full NeuCube STAM-EEG classification model was trained on all 60 samples and 14 variables. The parameter settings of the STAM-EEG NeuCube model are shown in Table 1 (for explanation, see [6,11]).

Parameter values of a NeuCube model influence a great deal the performance of the model. There are several ways to deal with this problem. If there is domain knowledge related to the data and the problem in hand, that would instruct some of the parameter values, that will be a first step. Different combinations of the values of other parameters can be experimented with using either a Grid search or evolutionary computation methods, with an objective function to reduce the classification error [9]. The parameter values in Table 1 are default parameters for a NeuCube model with the single aim to demonstrate the methods and they can be further optimized.

The fully trained NeuCube STAM-EEG classification model was first analyzed for connectivity and neuronal spiking activity (Figure 4a–c) and then tested for different accuracies (Table 2, Table 3, Table 4 and Table 5), also using a newly introduced here Retained Memory Accuracy (RMA), calculated using Equation (6) below:(6)$$RMA=\frac{Ar}{Af}$$
where *Af* is the classification accuracy of the full STAM model, and *Ar* is the retained accuracy of the model, validated received association or generalization on shorter time windows of data or less number of variables.

Table 2 tested the temporal *association* classification accuracy of the model. Table 3 shows the temporal *generalization* accuracy when 50% of the data was used for training the full model and 50% for validation. It showed that RMA = 1 when the model was validated on T1 time of 95% and RMA = 0.95 for 80% of the time of the data used. Similar experiments are shown in Table 4 and Table 5 for evaluating the spatial association and generalization accuracy of the model correspondingly. When one of the input variables (T7, ranked lowest according to Figure 4c) was removed when the model was validated, the RMA was still very high.

The proposed STAM-EEG classification method is illustrated here on a simple EEG problem, but its applicability is much wider across various studies involving EEG or ECoG data. A large STAM-EEG model can be developed for a particular problem. This model can be validated for its temporal and spatial association and generalization accuracy on a particular subset of EEG channels, measured at shorter times. If the validation accuracies are acceptable, then the model can be successfully used on smaller EEG data. This method can be used for the early detection of brain events in an online mode, using only a shorter time of activity of a small number of channels. Further applicability of the proposed STAM-EEG classification method is discussed in Section 6.

## 5. STAM-fMRI for Classification

### 5.1. The Proposed STAM-fMRI Classification Method


i.Defining the spatial and temporal components of the fMRI data for the classification task, e.g., fMRI voxels and the time series measurement.ii.Designing a SNNcube that is structured according to a brain template suitable for the fMRI data. This could be a direct mapping of the fMRI voxel coordinates or transforming the voxel coordinates from the fMRI image to another template, such as Talairach, MNI, etc. [20] (Figure 5a).iii.Selecting voxel features/variables K from the full set of voxels (Figure 5b) and defining their mapping as input neurons in the 3D SNNcube. (Figure 5c).iv.Encode data and train a NeuCube model to classify a complete spatio-temporal fMRI data, having K variables as inputs measured over time T.v.Analyse the model through connectivity and spiking activity analysis around the input voxels (Table 3).vi.Recall the STAM-fMRI model on the same data and same variables but measured over time T1 < T to calculate the classification temporal association accuracy.vii.Recall the STAM-fMRI on K1 < K EEG channels to evaluate the classification spatial association accuracy.viii.Recall the model on the same variables, measured over time T or T1 < T on new data to calculate the classification temporal generalization accuracy.ix.Recall the NeuCube model on K1 < K variables to evaluate the classification spatial generalization accuracy using a new fMRI dataset.x.wRank and evaluate the K1 fMRI features/variables as potential classification biomarkers (Section 5.5).


### 5.2. STAM-fMRI for Classification of Experimental fMRI Data

The experimental fMRI data set used here was originally collected by Marcel Just and his colleagues at Carnegie Mellon University’s Center for Cognitive Brain Imaging (CCBI) [21]. The fMRI recorded 5062 voxels from the whole brain volume while a subject was performing a cognitive reading task. There were two categories of sentences (affirmative and negative), each remaining on the screen for 8 s corresponding to 16 measured brain images. There were a total number of 40 sentences.

A full STAM-fMRI model was developed for the classification of fMRI samples into two classes (class 1: affirmative sentence and class 2: negative sentence). Signal to noise ratio (SNR) feature selection method was applied to the fMRI data to select vital fMRI variables with a high power of discrimination between the defined classes. As shown in Figure 5b, we selected 20 top important voxels that had SNR values higher than the 0.4 threshold. These 20 fMRI features are used as input variables to train the STAM-fMRI model for classification.

Figure 5a shows how 5062 voxel coordinates were mapped into a 3-dimensional SNNcube. Table 6 shows the brain region of interest (RoI) associated with these top-20 selected fMRI features and the evolved connectivity in the 3D SNN STAM model around the input features, as follows: LT (3), LOPER (3), LIPL (1), LDLPFC (6), RT (2), CALC (1), LSGA (1), RDLPFC (1), RSGA (1), RIT (1). The full names of the areas are: left temporal lobe (LT); left opercularis (LOPER); left inferior parietal lobule (LIPL); left dorsolateral prefrontal cortex (LDLPFC) and right dorsolateral prefrontal cortex (RDLPFC); calcarine sulcus (CALC); right supramarginal gyrus (RSGA).

The training classification accuracy of the full STAM-EEG classification model was 100% (Figure 5c,d) and the associative temporal and spatial testing accuracy of the model was further tested and presented below.

Figure 6a presents three snapshots of deep learning of eight-second fMRI data in a SNNcube when a subject was reading a negative sentence (time in seconds). Figure 6b captures the internal structural pattern, represented as spatio-temporal connectivity in the SNN model trained with eight-second fMRI data streams. The corresponding functional pattern is illustrated in Figure 6c as a sequence of spiking activity of clusters of neurons in a trained SNNcube. The internal functional dimensionality of the SNN model shows that while the subject was reading a negative sentence, the activated cognitive functions were initiated from the Spatial Visual Processing function. Then it was followed by the Executive functions, including decision-making and working memory. From there, the Logical and Emotional Attention functions were involved. Finally, the Emotional Memory formation and Perception functions were evoked.

### 5.3. The Full STAM-fMRI Classification Model Is Recalled on Partial Temporal fMRI Data

Here the trained full STAM-fMRI model in Section 5.2 was recalled on 70% and 50% of the time length of the same data used for the training (Figure 7).

The classification temporal association accuracy for both experiments was 100%. Using less than 50% of the time series resulted in an accuracy of less than 100%.

### 5.4. Testing the Full STAM-fMRI Model on a Smaller Portion of the Spatial Information (a Smaller Number of fMRI Variables/Features)

Here, the STAM-fMRI model from Section 5.2, trained on 20 features, was validated only on 18 of them, by removing the last two from the SNR ranking (Figure 5b). The spatial classification association accuracy was again 100% (Figure 8). The accuracy decreases when less than 18 input variables are used.

### 5.5. Potential Bio-Marker Discovery from the STAM-fMRI

A fully trained STAM-fMRI classification model can be analyzed in terms of the most activated brain regions related to reading affirmative and negative sentences. Figure 9 shows the distribution of the average connection weights around the input features located in the left and right hemispheres of the trained SNN models related to reading different sentences.

### 5.6. STAM for Longitudinal MRI Neuroimaging

STAM systems can be developed also for longitudinal MRI data (STAM-longMRI), such as the one used in [22], where 6 years of MRI data has been modeled to predict dementia and AD in 2 and 4 years ahead from a large cohort of data. A STAM-longMRI system can be trained on the full length of longitudinal MRI data and used to be recalled in a shorter time for early prediction of future events.

## 6. Discussions, Conclusions, and Directions for Further Research

### 6.1. Potential Applications of the Proposed STAM-NI Classification Methods

The potential applications of the STAM-NI classification methods proposed here become evident in various fields, including post-stroke recovery prediction, early diagnosis, and prognosis of mild cognitive impairment (MCI) and Alzheimer’s disease (AD), as well as depression and other mental health conditions. These applications can be NI techniques such as EEG and fMRI to analyze spatio-temporal patterns of brain activity and make accurate and early predictions or classifications.

One notable application is in post-stroke recovery prediction. By training a STAM model on NI data collected from stroke patients, the model can learn the spatio-temporal patterns associated with successful recovery. Subsequently, the model can be recalled using only partial NI variables or time points to predict the recovery trajectory of the same patient or a new stroke patient. This capability can assist clinicians in personalized treatment planning and rehabilitation strategies [23,24].

Another application lies in the early diagnosis and prognosis of MCI and Alzheimer’s disease. By training a STAM model on longitudinal NI data, such as EEG and fMRI recordings, from individuals with and without MCI/AD, the model can learn the complex spatio-temporal patterns indicative of disease progression. The model can then be utilized to classify new individuals based on their NI data, enabling early detection and intervention for improved patient outcomes [25,26].

Depression is another mental health condition that can benefit from the STAM framework. By training a STAM-NI model on NI data, such as resting-state fMRI, from individuals with depression, it can capture the spatio-temporal associations related to the disorder. This trained model can subsequently be used to classify new individuals as either depressed or non-depressed based on their NI data, aiding in early diagnosis and treatment planning [27].

Furthermore, the STAM systems hold potential for applications in neurodevelopmental disorders, such as autism spectrum disorder (ASD). By training a STAM model on EEG data, it can identify distinctive spatio-temporal patterns associated with ASD, contributing to early diagnosis and intervention [28]. Similarly, the framework can be applied to investigate brain disorders related to aging, such as Parkinson’s disease or age-related cognitive decline [29].

By incorporating multimodal spatio-temporal data, including clinical, genetic, cognitive, and demographic information, during the training phase, a STAM model can enable comprehensive analyses. This integration of multiple modalities aims to enhance the model’s ability to make accurate predictions or classifications, even when only a subset of the modalities is available for recall. Such a capability can provide valuable insights for personalized medicine, treatment planning, and patient management [30].

One challenge in the STAM system design is how it can effectively associate different data modalities during learning, enabling successful recall even when only one modality is available. For instance, can a STAM model learn brain data from synesthetic subjects who experience auditory sensations when they see colors? Addressing this challenge requires leveraging prior knowledge about brain structural and functional pathways, as well as stimuli data and corresponding spatio-temporal data from subjects. Current understanding of structural connectivity and functional pathways during perception can be utilized to initialize the connectivity of the SNN Cube before training [31,32,33].

Another open question pertains to how sound, image, and brain response data (e.g., EEG, fMRI) can be inputted as associated spatio-temporal patterns into dedicated groups of neurons. This concept aligns with the principles employed in neuroprosthetics, where stimulus signals are delivered to specific brain regions to compensate for damage, effectively “skipping” damaged areas [34,35]. Experiments conducted using the STAM-NI framework have the potential to provide insights and ideas for the development of new types of neuroprosthetics that leverage spatio-temporal associations in neural activity.

Wider applications of the proposed STAM models can be anticipated, such as predicting air pollution [36] with the use of neuromorphic hardware [37,38,39,40].

### 6.2. Future Development and Challenges of the STAM-NI Methods

STAM-NI methods can be developed in the future to address the following challenges:-Developing new functions in the NeuCube SNN, enabling a better STAM system design that are inspired by neurogenetic [41] and brain cognition [42,43,44] and also enhancing already existing SNN models for transfer learning and knowledge discovery [45,46,47].-Normalizing or/and harmonizing NI data across various data sources [48]. Establishing an effective “mapping” between training variables and synchronized time units will be crucial.-Implementation of STAM models on neuromorphic microchips, consuming much less energy and being implantable for online adaptive learning and control [37,38,39,40,49]. The choice of a hardware platform for the implementation of a practical STAM system would depend on the specific task requirements.-STAM-NI, which works under different temporal conditions, e.g., with data collected at varying intervals. At present the same time unit is used for training and for (e.g., milliseconds, seconds, etc.). If the recall data is measured in different time intervals, we can apply interpolation between the data points so that they will match the training temporal units. Such data interpolation has been successfully used in brain data analysis using the NeuCube SNN [22].-STAM-NI, for different spatial settings. At this stage, we have explored the model when data for training and recall are in the same spatial setting and same context. We can explore further the ability of the model for incremental learning of new variables, that can be mapped spatially. In this case, the network of connections in the 3D SNN will form new clusters that connect spatially the new variables and may also develop links with the “old” variables.-STAM-NI, which accounts for the variability of the variables themselves. In real-world scenarios, variables may have different characteristics, and their relationships may evolve. We will consider how a STAM framework performs with diverse types of variables, including those with different temporal dynamics and spatial distributions.-In conclusion, the proposed STAM-NI classification framework and its specific models STAM-EEG and STAM-fMRI are not aimed to substitute existing methods and systems for NI data analyses. Rather, they are extending their functionality for better NI data modeling, data understanding, and early event diagnosis and prognosis.

## Figures and Tables

**Figure 1 bioengineering-10-01341-f001:**
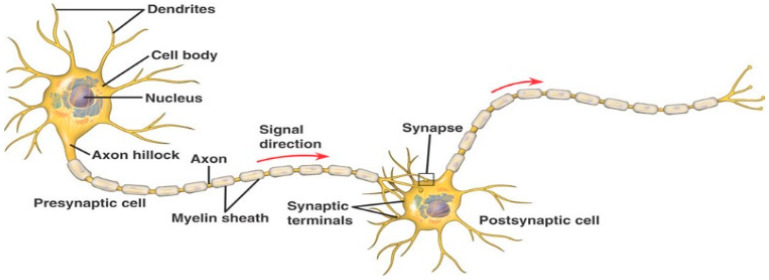
Learning in SNN relates to changes in the connection weights between two spatially located spiking neurons over time so that both “time” and “space” are learned in the spatially distributed connections (http://en.m.wikipedia.org/wiki/neuron, accessed on 13 November 2023).

**Figure 2 bioengineering-10-01341-f002:**
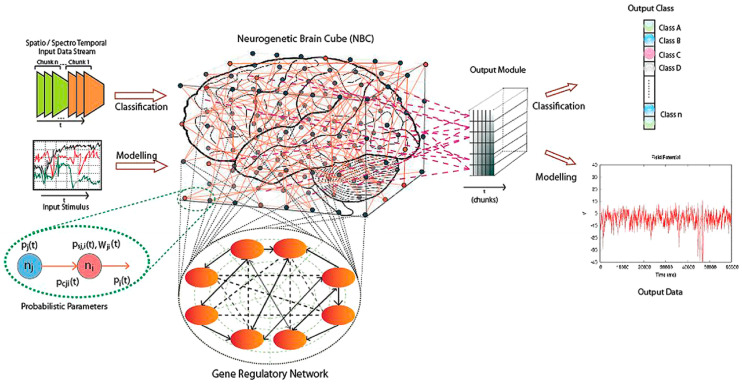
The NeuCube brain-inspired SNN architecture (from [6], (©Elsevier, reproduced with permission from Kasabov, N., NeuCube: A spiking neural network architecture for mapping, learning and understanding of spatio-temporal brain data, Neural Networks, vol. 52, 2014)).

**Figure 3 bioengineering-10-01341-f003:**
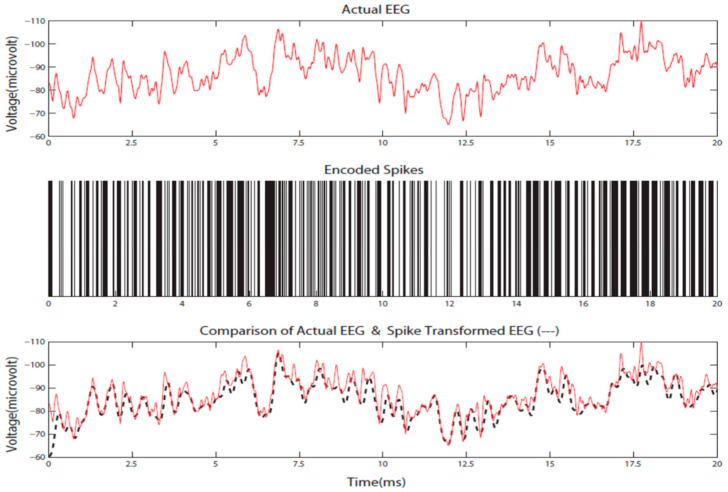
Original EEG signal (**top**), encoded into spike sequence (**middle**) and a reconstruction of the signal from the spike sequence, back to real values (**bottom**) (from [9], (©Springer-Nature 2019, reproduced with permission from Kasabov, N., Time-Space, Spiking Neural Networks and Brain-Inspired Artificial Intelligence, 2019)).

**Figure 4 bioengineering-10-01341-f004:**
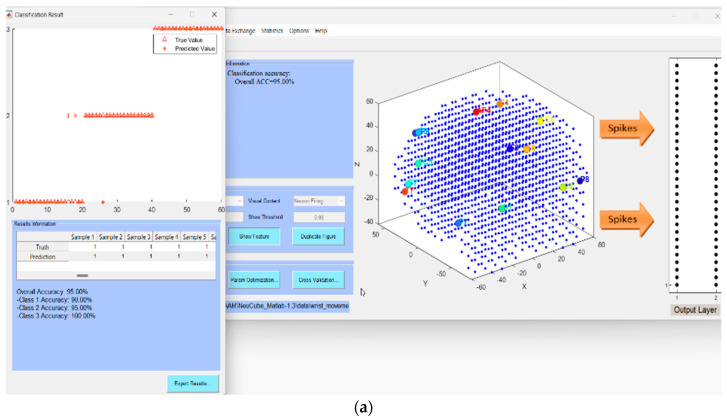
(**a**) Training the NeuCube STAM-EEG model on full data (60 EEG samples) and validating it on T1 = 80% of the time of the data (see Table 2). Different input neurons, representing corresponding EEG channels, are presented in different colors; (**b**) Post-training neuronal connectivity and cluster formations. (**c**) (**Left**): The size of the segments represents the spiking activity of the corresponding input neuron to an EEG channel; the largest the section, the higher the impact this channel has on the model; (**Right**): EEG electrode layout.

**Figure 5 bioengineering-10-01341-f005:**
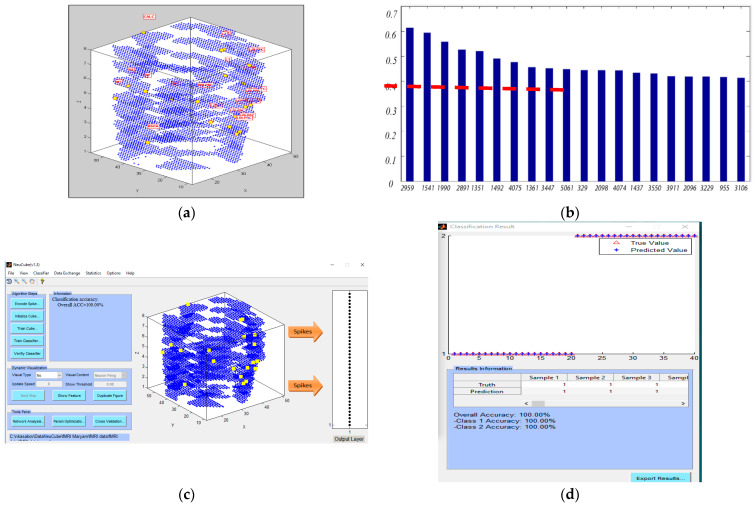
(**a**) Mapping of the 5062 fMRI voxels into a 3D SNN model of the NeuCube framework; (**b**) selecting top-20 voxels as input variables using SNR ranking (on the *y*-axis) of top voxels (on the *x*-axis) related to the affirmative versus negative sentences. The top features are selected according to their SNR values that were greater than a threshold = 0.4. (**c**) a full STAM-fMRI model implemented in NeuCube trained and tested on 100% of the data using all 20 features; (**d**) its training accuracy is 100%, but the validation association and generalization accuracies are further tested below.

**Figure 6 bioengineering-10-01341-f006:**
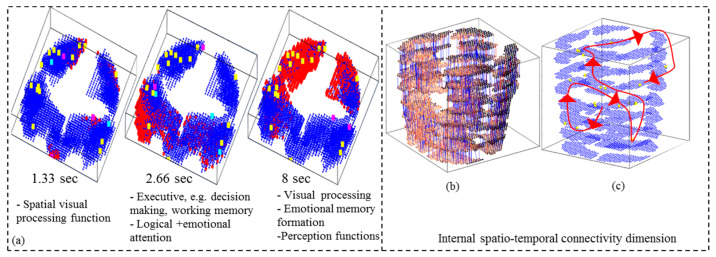
(**a**) Three snapshots of learning of 8-s fMRI data in a STAM-fMRI model when a subject is reading a negative sentence (time in seconds); Positive connections are colored in blue and negative connections in red. (**b**) Internal structural pattern represented as spatio-temporal connectivity in the SNN model trained with 8-s fMRI data stream; (**c**) A functional pattern represented as a sequence of spiking activity of clusters of spiking neurons in a trained SNN model. The arrows show the order of activation of different spatially distributed neuronal areas after fMRI data is presented to an already trained SNNcube.

**Figure 7 bioengineering-10-01341-f007:**
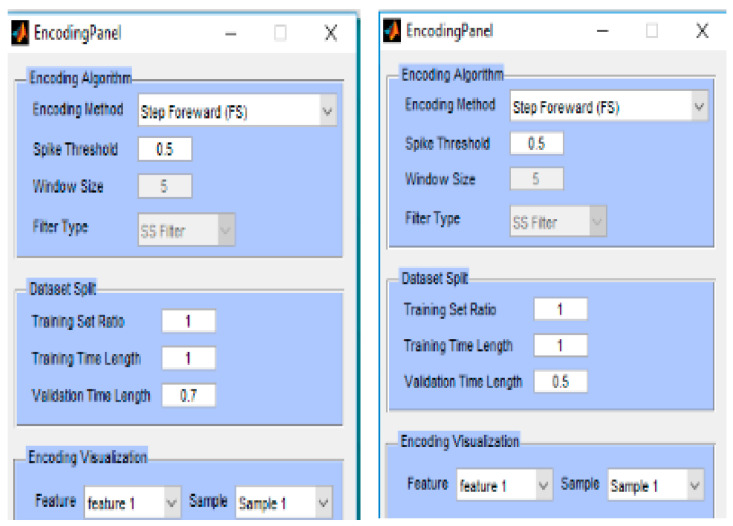
Parameters for spike encoding and validation of the STAM-fMRI model from Section 5.2. Left panel: For validation, only 70% (0.7) from the initial time points of the fMRI samples, equaled to 5.6-s data, are used, rather than using 8 s of the data for training the full model. Right panel: The model is tested/validated only on 50% of the temporal length (4 s) of the training data. The classification temporal association accuracy for both experiments is 100%. Using less than 50% of the time series results in an accuracy of less than 100%.

**Figure 8 bioengineering-10-01341-f008:**
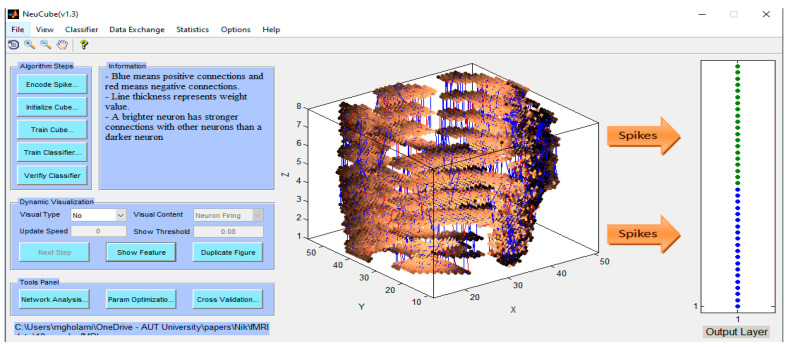
Classification spatial association accuracy is 100% when 18 input features are used. The panel at the write shows the correct classification of all input fMRI samples in class 1 (in green) and class 2 (in blue).

**Figure 9 bioengineering-10-01341-f009:**
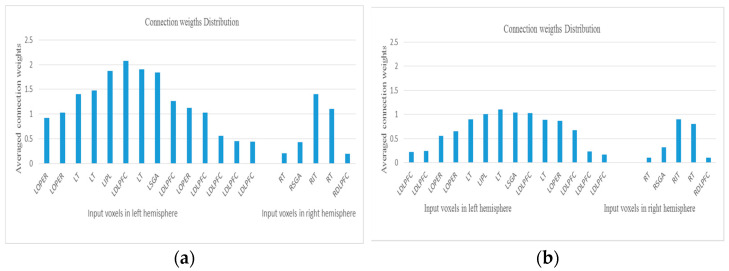
Distribution of the average connection weights around the input voxels located in the left and right hemispheres of the trained SNN models related to negative sentences (in (**a**)) and affirmative sentences (in (**b**)). The dominant voxels for the discrimination of the negative from the affirmative sentences are LDLPFC, LIPL, LT, and LSGA.

**Table 1 bioengineering-10-01341-t001:** STAM-EEG Parameter Settings of a NeuCube model.

Dataset Information	Encoding Method and Parameters	NeuCube Model	STDP Parameters	deSNNs Classifier Parameters
sample number: 60, feature number: 14 channels, time length: 128, class number: 3.	encoding method: Step Forward (SF) spike threshold: 0.5, window size: 5, filter type: SS.	number of neurons: 1471, brain template: Talairach, neuron model: LIF.	potential leak rate: 0.002, STDP rate: 0.01, firing threshold: 0.5, training iteration: 1, refractory time: 6, LDC probability: 0.	mod: 0.8, drift: 0.005, K: 1, sigma: 1.

**Table 2 bioengineering-10-01341-t002:** Temporal association accuracy of the STAM-EEG model from Figure 4a–c.

Time T1 (% of the Full Time for Training)	Number of Input Variables	Training/Validation % of Data Samples	Temporal Association Accuracy	RMA
100% (full)	14 (100%)	100/100	100%	1
95%	14 (100%)	100/100	100%	1
90%	14 (100%)	100/100	98%	0.98
80%	14 (100%)	100/100	95%	0.95

**Table 3 bioengineering-10-01341-t003:** Temporal generalization accuracy of the STAM-EEG model from Figure 4a–c.

Time T1 (% of the Full Time)	Number of Input Variables Used	Training/Validation % of Data Samples	Temporal Generalization Accuracy	RMA
100% (full)	14 (100%)	50/50	80%	1
95%	14 (100%)	50/50	80%	1
90%	14 (100%)	50/50	76%	0.95

**Table 4 bioengineering-10-01341-t004:** Spatial association accuracy of the STAM-EEG model from Figure 4a–c when feature T7 was removed.

Time T1 (% of the Full Time)	Number of Input Variables	Training/Validation % of Data Samples	Spatial Association Accuracy	RMA
100% (full)	14 (100%)	100/100	100%	1
100% (full)	13 (93%)	100/100	100%	1
95%	13 (93%)	100/100	100%	1
90%	13 (93%)	100/100	86%	0.86

**Table 5 bioengineering-10-01341-t005:** Spatial generalization accuracy of the STAM-EEG model from Figure 4a–c when feature T7 was removed.

Time T1 (% of the Full Time)	Number of Input Variables	Training/Validation % of Data Samples	Temporal Association Accuracy	RMA
100% (full)	14 (100%)	50/50	80%	1
100% (full)	13 (93%)	50/50	100%	1
95%	13 (93%)	50/50	80%	1
90%	13 (93%)	50/50	76%	0.95

**Table 6 bioengineering-10-01341-t006:** The level of the evolved connectivity of each input feature neuron, representing a local brain area when a person is reading a negative (Neg) vs. affirmative (Aff) sentence can be used for feature selection and bio-marker discovery; the higher the value, the more important the input feature is.

Area	LT	LOPER	LIPL	LOPER	LDLPFC	LOPER	LT	LDLPFC	RT	CALC	
Neg	1.4	0.92	1.87	1.03	2.08	1.12	1.48	0.44	0.2	0.89	
Aff	0.9	0.56	1.01	0.87	1.03	0.65	0.89	0.23	0.1	0.43	
area	LSGA	LDLPFC	LT	LDLPFC	RT	LDLPFC	LDLPFC	RDLPFC	RSGA	RIT	Avg
Neg	1.84	1.03	1.9	0.45	1.1	1.26	0.56	0.19	0.43	1.4	1.7
Aff	1.04	0.68	1.1	0.17	0.8	0.24	0.22	0.11	0.32	0.9	0.6

## Data Availability

All data is publicly available as described in the paper.

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
