# Peer review of "Brain-Inspired Spatio-Temporal Associative Memories for Neuroimaging Data Classification: EEG and fMRI"

_bioengineering, 2023, doi:10.3390/bioengineering10121341_

Round 1
Reviewer 1 Report
Comments and Suggestions for Authors
The authors present their work on a classification model of fMRI and EEG. The manuscript is well written and clear. The quality of the language and presentation is sufficient.
I am a bit puzzled why one part of the manuscript is written in red, other than that I have no issues with the manuscript.
Author Response
Many thanks for the positive and encouraging review of the new method that we would like to publish first in this journal.
The red colour in the text indicated some initial changes after the preliminary review. In the latest version of the paper all text will be in black.
Reviewer 2 Report
Comments and Suggestions for Authors
In the manuscript, the application of specific AI platform has been exploited utilizing EEG and fMRI data. However, there are several issues that should be further processed and analyzed in order the methodology and the whole system to be correctly evaluated. Some of the critical concerns are:
The generalization ability of the system – it seems that is quite poor.
Multiple sources and various data types should be utilized in both training and testing processes.
Most of the techniques that are introduced are only mentioned without any technical description of the methodology that is on their background. The selection and the characteristics of the specific dataset, the variety and the final utilization of the parameters as well as their influences to the overall systems performance should be presented and clarified.
Although the overall performance hasn’t been presented clearly, the association accuracy of the technique appeared quite high (100%) something that rises overtraining issues and should be investigated.
Author Response
Q: In the manuscript, the application of specific AI platform has been exploited utilizing EEG and fMRI data. However, there are several issues that should be further processed and analyzed in order the methodology and the whole system to be correctly evaluated. Some of the critical concerns are:
A: The generalization ability of the system – it seems that is quite poor.
Thank you for the comment. For testing the generalisation temporal and spatial accuracy of the STAM-EEG model, more experiments are conducted when 50% of the EEG data used for training a STAM-EEG model and 50% new data for testing. For the temporal generalization accuracy 90% and 80% of the original data times points are used. For testing the spatial generalisation accuracy 13 features are used instead of 14, on 50% of new data. Results are shown in new tables 2, 3,4 and 5.
Q: Multiple sources and various data types should be utilized in both training and testing processes.
A: The paper deals with two types of neuroimaging data, as shown in the title. We agree that further testing of the proposed approach needs to be done, and we have indicated now in the discussion section.
Q: Most of the techniques that are introduced are only mentioned without any technical description of the methodology that is on their background. The selection and the characteristics of the specific dataset, the variety and the final utilization of the parameters as well as their influences to the overall systems performance should be presented and clarified.
A: The revised version adds a detailed description of all parameters used and discusses the parameters' influence on the systems' performance (page 6).
Q: Although the overall performance hasn't been presented clearly, the association accuracy of the technique appeared quite high (100%) something that rises overtraining issues and should be investigated.
A: The high temporal association accuracy (100%), when only 70% and 50% of the fMRI data time points are used to recall a trained model on 100% of the time points (8 sec) of the 20 fMRI features, shows that the STAM-fMRI model has a good temporal associative accuracy up to 50% of reduced time of the features. It is also mentioned that when less than 50% of the time points are used, the association accuracy goes down.
From the authors: Many thanks to the reviewer for the insightful comments.
16/10/2023
Reviewer 3 Report
Comments and Suggestions for Authors
The paper proposes a brain-inspired spatio-temporal associative memory (STAM) framework that utilizes the NeuCube brain-inspired spiking neural network to analyze neuroimaging data, specifically EEG and fMRI. The study demonstrates that the STAM framework can accurately classify and recall information from partial time series and variables, opening possibilities for the development of multimodal classification systems and brain diagnostic/prognostic marker discovery using spatio-temporal neuroimaging data. Overall, the paper is well-written. Below are my comments and suggestions.
· The study evaluates the temporal association accuracy and spatial association accuracy of the STAM framework but does not provide a comprehensive analysis of its performance in different settings, labs, and clinics that may vary in terms of variables, time of data collection, and other parameters such as
o The study assesses how well the STAM framework can establish temporal associations between events or variables. This suggests that the framework might be effective in capturing how certain variables change over time. However, it doesn't delve deeper into how the framework performs under different temporal conditions. For instance, does it perform just as effectively when dealing with both short-term and long-term data collections? And does it sustain its accuracy when we use it with data collected at varying intervals, like on a daily, weekly, or monthly basis? Such variations in data collection frequency can impact the framework's effectiveness.
o Similarly, the study examines the framework's ability to establish spatial associations. This implies that the STAM framework might excel at recognizing relationships between variables in a specific spatial context. However, it doesn't provide insights into how the framework performs across different spatial settings. Is the framework equally accurate in different geographic locations or within different spatial scales (e.g., urban vs. rural)? These variations can significantly affect the framework's applicability and accuracy.
o The study may not account for the variability of the variables themselves. In real-world scenarios, variables may have different characteristics, and their relationships may evolve over time. A comprehensive analysis should consider how the STAM framework performs with diverse types of variables, including those with different temporal dynamics and spatial distributions.
· The paper does not discuss the potential limitations or challenges associated with the implementation and practical use of the STAM framework in real-world scenarios. The research paper misses out on a crucial aspect by not putting the STAM framework head-to-head with other methods or approaches used in neuroimaging data analysis. This kind of comparison could really help us understand where STAM shines and where it might have some shortcomings compared to existing techniques.
· It doesn't delve into the nitty-gritty details of how much computing power and resources the STAM framework demands. This is quite significant because knowing these requirements is vital for assessing how feasible and scalable it is in real-world applications.
Author Response
The paper proposes a brain-inspired spatio-temporal associative memory (STAM) framework that utilizes the NeuCube brain-inspired spiking neural network to analyze neuroimaging data, specifically EEG and fMRI. The study demonstrates that the STAM framework can accurately classify and recall information from partial time series and variables, opening possibilities for the development of multimodal classification systems and brain diagnostic/prognostic marker discovery using spatio-temporal neuroimaging data. Overall, the paper is well-written. Below are my comments and suggestions.
Q: The study evaluates the temporal association accuracy and spatial association accuracy of the STAM framework but does not provide a comprehensive analysis of its performance in different settings, labs, and clinics that may vary in terms of variables, time of data collection, and other parameters
A: Thank you for your valuable feedback. We appreciate your insightful comment regarding the need for comprehensive performance analysis in diverse settings, labs, and clinics. As this paper represents the inaugural exploration of the STAM-NI framework for neuroimaging data, we acknowledge the importance of normalizing or harmonizing the method for application across various data sources. Establishing an effective "mapping" between training variables and synchronized time units will be crucial. We have incorporated these considerations into our conclusion to highlight the significance of this aspect for future applications of the STAM framework (a new reference [53]).
Q: The study assesses how well the STAM framework can establish temporal associations between events or variables. This suggests that the framework might be effective in capturing how certain variables change over time. However, it doesn't delve deeper into how the framework performs under different temporal conditions. For instance, does it perform just as effectively when dealing with both short-term and long-term data collections? And does it sustain its accuracy when we use it with data collected at varying intervals, like on a daily, weekly, or monthly basis? Such variations in data collection frequency can impact the framework's effectiveness.
A: Thank you for the comment and the very good point made. We suggest that the time unit used for training and recall is the same (e.g., milliseconds, second, etc.). If the recall data is measured in different time intervals, we can apply interpolation between the data points so that they will match the training temporal units. Such data interpolation has been successfully used in brain data analysis using the NeuCube SNN. References showing such brain data interpolation are cited as [22,39]. This text is included in the discussion part.
Q: Similarly, the study examines the framework's ability to establish spatial associations. This implies that the STAM framework might excel at recognizing relationships between variables in a specific spatial context. However, it doesn't provide insights into how the framework performs across different spatial settings. Is the framework equally accurate in different geographic locations or within different spatial scales (e.g., urban vs. rural)? These variations can significantly affect the framework's applicability and accuracy.
A: This is a perfect question. Thank you! At this stage, we have explored the model when data for training and recall are in the same context (e.g. urban or rural). Still, the method allows for incremental learning of new variables that can be mapped spatially according to the used algorithm and further training on new data . In this case the network of connections in the 3D SNN will form new clusters that connect spatially the new variables but may also show some links with the “old” variables that are used for training and were of a different context. This option needs to be further investigated and this is mentioned also in the discussion section.
Q: The study may not account for the variability of the variables themselves. In real-world scenarios, variables may have different characteristics, and their relationships may evolve over time. A comprehensive analysis should consider how the STAM framework performs with diverse types of variables, including those with different temporal dynamics and spatial distributions.
A: This is a very good suggestion for future work. Many thanks. At present, we can develop an ensemble of STAM models, each dealing with their own temporal and spatial distributions of variables and data and merge them based on an objective function. Another option is to build a complex hierarchical STAM, where lower level clusters are connected to higher level variables that are functioning at different time scales. We feel encouraged by the reviewer to extend our research in these directions. We will be honoured to invite the reviewer to join us in such future research, which is a cutting-edge one indeed.
Q: The paper does not discuss the potential limitations or challenges associated with the implementation and practical use of the STAM framework in real-world scenarios. The research paper misses out on a crucial aspect by not putting the STAM framework head-to-head with other methods or approaches used in neuroimaging data analysis. This kind of comparison could really help us understand where STAM shines and where it might have some shortcomings compared to existing techniques.
A: Thanks you for the comments. We have added a paragraph about the advantages of STAM when compared with traditional NI data analysis (the last paragraph in section 6). In conclusion, the proposed STAM-NI framework and its specific models STAM-EEG and STAM-fMRI are not aimed to substitute existing methods and systems for NI data analyses. Rather they are extending their functionality for a better NI data modelling, data understanding and for early diagnosis and prognosis . This text is added to the discussion part.
Q: It doesn't delve into the nitty-gritty details of how much computing power and resources the STAM framework demands. This is quite significant because knowing these requirements is vital for assessing how feasible and scalable it is in real-world applications.
A: Thanks for the question. A STAM system is based on a SNN and can be implemented on massively parallel and a high-speed neuromorphic hardware, such as the Manchester SpiNNaker, the Intel Loihi or other that consume much less power than traditional computers (ref [ 42-45] and a new reference [54]). The hardware platform used for a specific practical implementation will depend of the task specific requirements.
Round 2
Reviewer 2 Report
Comments and Suggestions for AuthorsΑll questions and comments have been satisfactorily answered.